# A Goal-oriented Neural Conversation Model by Self-Play

## Abstract

Building chatbots that can accomplish goals such as booking a flight ticket is an unsolved problem in natural language understanding. Much progress has been made to build conversation models using techniques such as sequence2sequence modeling. One challenge in applying such techniques to building goal-oriented conversation models is that maximum likelihood-based models are not optimized toward accomplishing goals. Recently, many methods have been proposed to address this issue by optimizing a reward that contains task status or outcome. However, adding the reward optimization on the fly usually provides little guidance for language construction and the conversation model soon becomes decoupled from the language model. In this paper, we propose a new setting in goal-oriented dialogue system to tighten the gap between these two aspects by enforcing model level information isolation on individual models between two agents. Language construction now becomes an important part in reward optimization since it is the only way information can be exchanged. We experimented our models using self-play and results showed that our method not only beat the baseline sequence2sequence model in rewards but can also generate human-readable meaningful conversations of comparable quality.

## 1 Introduction

Building chatbots that can naturally interact with human users has long been an important challenge in artificial intelligence and computer science (Turing, 1950). Recently, there is growing interest in applying end-to-end neural networks to this task with promising results (Vinyals & Le, 2015; Shang et al., 2015; Sordoni et al., 2015). A compelling aspect of these models is that they require fewer hand-crafted rules compared to traditional models. Their success is however limited to conversations with very few turns and without any goals (also known as "chitchat").

The goal of this work is to build goal-oriented conversational models. Here we use "goal-oriented" to mean that the model must accomplish a particular desired goal in the dialogue. Depending on the nature of the task, conversations can be as simple as few-round dialogues such as resetting passwords or it can involve back and forth investigations in the case of travel recommendation and IT support. Different from chitchat based conversation models, whose goal is to generate response without task restrictions, goal-oriented models will have to direct the conversations in a way that facilitates the progress of the task. For example in the case of flight booking, customers are only interested in moving the conversation forward if the flight recommendations meet their expectations. Similarly, agents are only supposed to make responses that will resolve customers' requests.

Building goal-oriented conversational models presents a fresh challenge to neural network-based conversational models because their success in chitchat dialogues can not be easily transferred into the world of goal-oriented dialogues. Firstly, chitchat models trend to remember the exact settings of the context-response pairs. Due to the high variance of deep models, slight changes in the context such as cities, time or names will likely change the response completely. Although one can provide more training data to cover different combinations of these pieces of information, acquiring dialogue data for the exhaustive set of conditions is difficult or in many cases infeasible. Secondly, the fact that most chitchat models optimize the likelihood of the utterances makes it hard for them to generate responses that are less likely in general but are appropriate given the context of the task. The progress of the dialogue can easily get lost during the conversation and agents might

not be able to reach to the optimal conclusion. And finally, while in most chitchat models, diversity of the responses is one of the key metrics, goal-oriented conversation models focus on robustness and reliability of the response especially in it's roles to guide the conversation.

To address these problems, in this paper we propose a two-party model for goal-oriented conversation with each sub-model being responsible for its own utterances. We define the goal oriented conversation problem to be a dialogue cooperation with two agents each having access to some pieces of hidden information that is only visible to itself. One of the agents is required to come up with a series of "action", which correctness relies solely on the understanding of the hidden information. Although the exact form of the hidden information is not visible, it can be interpreted using natural language and be exchanged to the other party. In order to achieve this, agents need to establish a protocol to talk and complete the task correctly.

The two-party architecture makes it feasible to conduct self-play between agents, which enables two conversation models to talk to itself without human supervision. Different from previous proposed self-play dialogue models such as the negotiation chatbot (Lewis et al., 2017), our setting enforces the isolation of information between the models of the two agents, which ensure the coupling between task reward and language models. And because hidden information is not directly visible, agents will need to guide and structure the conversation in a proper way in order to acquire the key pieces of information that is required to generate the correct actions. This process can be naturally strengthened using self-play with reinforcement learning.

Another benefit of the self-play model is the ability to utilize large scale un-supervised knowledge that is otherwise difficult to leverage. Training of dialogue models require lots of data but supervised conversation data are usually hard to acquire. Fortunately, it is usually easy to generate the initial conditions of the dialogue such as user restrictions available information in the database. Based on those initial settings, a rule based program is usually enough to generate action states, which constitutes a perfect reinforcement learning environment to estimate rewards. Using self-play to exchange hidden information, we can potentially leverage knowledge of a much larger scale and train a much more reliable chatbot.

To validate the performance of the model we first trained a supervised model based on fully supervised dialogue utterance, action states and hidden conditions. The supervised model is then used as a bootstrap to initialize a self-play training model based on initial conditions without supervised dialogues. We evaluated both models on a held-out self-play data sets and observed 37% performances improvements on average rewards for the agent learned from self-play.

## 2 PROBLEM SETTING AND MODEL ARCHITECTURE

Although the settings can be applied to a much broader scenario, here we consider a specific problem of booking flight tickets. In this problem, there exists a customer agent and a service agent, each having access to a pieces of hidden information that represents the customer's travel restriction and the available flights in the database. Customer agent have many different choices of making the request, such as book a flight, change a flight and cancel a flight. However, not all the requests can be fulfilled because the conditions might not be satisfied. Nevertheless, the goal of the service agent is to help the customer agent to the best of it can and make the correct "action states" to conclude the conversation. Depending on the restrictions of the customer and the available flights in the database, there could be different outcomes of the dialogue such as "booked" or "flight not found". The effectiveness of the conversation can then be evaluated based on whether those actions are correctly been recovered. The type of the tasks is chosen carefully so that exchanging and processing hidden information across models are necessary in order to come up with the optimal action state for the conversation.

Our model consists of two components that are mostly independent to each other. The only part that is shared between the components is the token embedding $E$, which transforms tokens into high dimensional representations and is optimized across the entire model. Structures for the two com-

ponents are similar except for the hidden knowledge encoder and the action state decoder. Figure 1 is an overview of the model structure.

Figure 1: Model Architecture

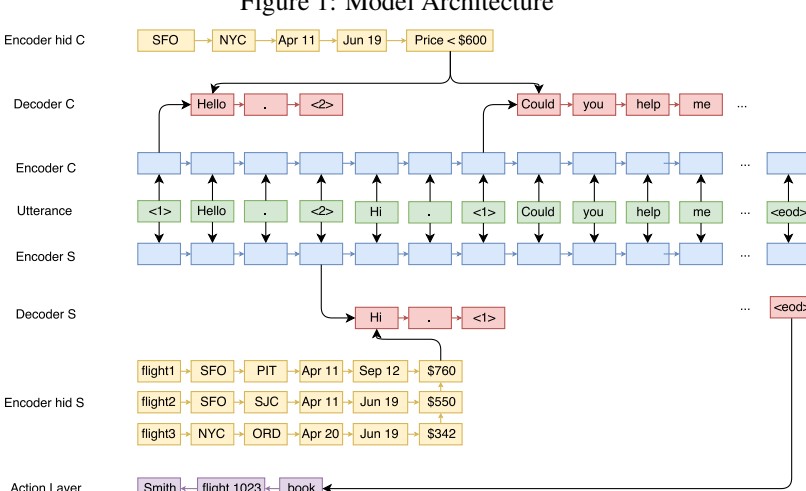

## 2.1 Training with Supervised Data

When dialogue utterance $U$ is present, we can train a supervised model for both the service agent and the customer agent. These models will be trained along with their hidden information, namely user restrictions $R$, database $K$ along with the optimal action state $A$.

**Encoding External Knowledge**    There are two types of hidden knowledge that we need to encode: user restriction $R$ and database $K$. The user restriction is sequence that can be encoded using a single GRU. We take the last hidden state to be its embedding representation, $h_R$. Encoding the hidden information of the agent can be a little bit more complicated as databases are usually consisted of two dimensional data. We first encode the row of the database entry using the first GRU. And then, we take the last hidden states of each row and feed them into a second GRU to acquire the embedding representation of the database $h_K$.

**Dialogue Encoder and Decoder.**    Dialogue context up to a certain position $T$ can be represented as a sequence of tokens $u_1, u_2, ..., u_T$. We use a special token such as "$<1>$" to represent the start of the turn for the customer agent. Similarly the start of the turn for the service agent can be defined accordingly such as "$<2>$". Obviously, the start turn symbol of one agent also marks the end of turn for the other agent. To mark the end of the dialogue, we use a spetial token such as "$<eod>$".

We use two multi-layered GRU unites, $GRU_{en}^c$ and $GRU_{en}^s$ with unit size $d$ to encode the dialogue context. This is done after a embedding look up is applied by multiplying the input with the embedding matrix $E$. Using two different encoders on the same input allows each agent to focus on the part of the context that are of most relevant to them. The encoded embedding is $h_t^c$ for customer agent and $h_t^s$ for the service agent.

$$h_t^{c/s} = GRU_{en}^{c/s}(h_{t-1}^{c/s}, Eu_t) \tag{1}$$

The encoder is followed by another multi-layered GRU with the same dimension $d$ to decode the concatenated embedding of both dialogue and the hidden knowledge. Again the customer and service agent has its own decoder. The output of the GRU are treated as logits and a token can be predicted by applying a max over its softmax of the logits. We initialize decoder GRU's cell state using the the concatenated embedding: $[h_T^c, h_R]$ for customer agent and $[h_T^s, h_K]$ for the service agent. Since the concatenation will change the size of the embedding, we use two transformation matrices to map it back to the embedding of size $d$.

Similar to standard seq2seq scheme, we feed initial start tokens into the decoder. The start token will depend on the agent that is talking. A utterance $u_t$ is then generated every time an input token is feeded into the decoder. After the new token is generated, it is treated as input for the next time step and feeded back into the decoder in order to generate the next token until the end of turn token is encountered.

$$o_t^{c/s} = GRU_{de}^{c/s}(o^{c/s_{t-1}}, \widehat{u}_t^{c/s}) \tag{2}$$

$$\widehat{u}_{t+1}^{c/s} = \operatorname{argmax} softmax(o_t^{c/s}) \tag{3}$$

**Action State Decoder.** After the dialogue encoder encounters an end of dialogue token, the conversation reaches to an end and the service agent is expected to conclude the conversation by generating a sequence of action states. Action states can be a single token or a sequence of them. In the flight booking scenario it will be three tokens: name of he person, flight recommendation and the action states. Here we utilizes a standard seq2seq decoder that is initialized by the last hidden state of the dialogue encoder $h_T^s$.

$$\widehat{A} = Seq2Seq(h_T^s) \tag{4}$$

**Optimization.** During the supervised learning, we optimize the model by considering loss from both the dialogue and the action states. We also optimize the two components of the model together in a single loss function. Let $\theta^s$ to represent the parameter set of service agent model and $\theta^c$ to the parameter set of the customer model, Equation 5 is the loss function of the supervised model. Since a token can only belong to one of the agents at a particular time t, we use a mask $m^c$ to mask out the loss terms that does not present on for the customer agent on step $t$ of sample $n$. That is, $m_{n,t}^c$ is 1 if token at position $t$ of sample $n$ belongs to the customer agent and 0 otherwise.

$$\begin{aligned}
\ell \ (\theta_{SL}) = \sum_n \sum_i & m_{n,t}^c log P(u_{n,i}|u_{n,1:i-1}|\theta_{SL}^c) + \\
& (1 - m_{n,t}^c) log P(u_{n,i}|u_{n,1:i-1}|\theta_{SL}^s) + \\
& \sum_n \sum_j log P(A_{n,j}|A_{n,1:j-1}|\theta_{SL}^s)
\end{aligned} \tag{5}$$

## 2.2 Training with Self-play

In the self-play setting, dialogue supervision is missing. We are left with the travel restrictions $R$, database $K$. We initialize the weights of the self-play model by using the trained model in the supervised setting. We then let two model instances to talk to each other, one with fixed parameters and the other with trainable parameters that will be updated based either on policy gradient and value function updates. A predicted dialogue $\widehat{u}$ is generated along with a predicted set of actions $\widehat{A}$. A reward terminal $G$ is then generated by feeding the predicted actions to the environment after the conversation is completed.

$$G = env(\widehat{A}|R, K) \tag{6}$$

Our algorithm is set to maximize this reward. During training, we also use a reward discount $\gamma$ to distribute contributions of the rewards to earlier time steps.

$$Ret_t = \gamma^{T-t}G \tag{7}$$

Table 1: Statistics of the data set

| num. samples | 500k | max. turn length | 19 |
|---|---|---|---|
| max. dialogue length | 298 | avg. turns | 15.3 |
| num. db | 13 | book rate | 55.34% |
| cancel | 1.18% | change | 0.8% |
| resv. not found | 18% | flight not found | 24.68% |
| num. vocab | 5,553 | Training Set | 150k |
| Evaluation Set | 25k | Inference Set | 25k |
| Self-Play Training Set | 250k | Self-play Eval Set | 50k |

### 2.2.1 DIALOGUE GENERATION

Different from the method we use during the supervised training in Equation 3, here we generate the dialogue by sampling from policy rather than taking the argmax from its logits.

$$P(\widehat{u'}_t^{c/s}) \sim exp^{(o^{c/s})} \tag{8}$$

**Value Network.** To reduce the variance of reinforcement learning, we first build a value network to estimate returns based on the current states (i.e., dialogue context and hidden knowledge). Each agent has its own value network with trainable parameters. We use a multi-layered GRU to encode the context $\widehat{u'}_t^{c/s}$ and acquire its last hidden state $z^{c/s}$. We make a doc product between $z^{c/s}$ and the hidden knowledge based encoded using the encoder (i.e., $h_K$ or $h_R$). The value function is estimated by optimizing a mean square error between the predicted return and the actual return from the environment.

$$v(u'^c_{1:t}) = z^c \cdot h_R$$
$$v(u'^s_{1:t}) = z^s \cdot h_K \tag{9}$$

**Policy Network.** We use the same structure of the supervised learning to be our policy network with weights initialized from the learned supervised learning model. We adopt REINFORCE Williams (1992) algorithm.

$$\nabla \ell \ (\theta_{RL}) = \sum_{u \sim p(u|\theta_{RL})} \mathbb{E}_u Ret_t \nabla logp(u_t|u_{1:t-1}; \theta_{RL}) \tag{10}$$

## 3 DATASET AND ENVIRONMENT

As reward evaluation requires an environment, we have built a simulator to generate all the data that is needed by this paper. The simulator simulates the process between customers who aims to change, cancel or book a flight and the service center agent who will fulfill the requests based on the randomly generated database. Figure 2 shows the workflow of the simulator. As one can see, the three user requests will ended up with one of the four terminal status. We record the terminal status as action states. Table 3 shows some statistics of the dataset. The dataset is consists of 500k samples that are divided into 5 different sets: 30% for supervised training, 5% for supervised evaluation, 5% for inference, 50% for self-play training and 10% for self-play evaluation. In the inference dataset, we randomly chose a stop point and feed only partial dialogue and the next sentence after that. In the two self-play datasets, we did not generate any dialogue utterances.

**Setting Generator.** The dialogue settings that contain user restrictions and flight database are randomly generated for each sample. Table 3 and Table 3 show the list of features and their possible values. Those two information are pretty much aligned. However, user restrictions contain much coarser information such as departure time= "morning/evening/all" and price < "500". The database contains the actual information in a much more detailed way. This means the conversation model

Figure 2: Simulator Workflows

Table 2: Features of User Restrictions

| no. | 1 | 2 | 3 | 4 | 5 | 6 |
|---|---|---|---|---|---|---|
| feat. | Dep. City | Ret. City | Dep. Month | Ret. Month | Dep. Day | Ret. Day |
| val. | cate. | cate. | 1-12 | 1-12 | 1-31 | 1-31 |
| feature | Dep. Time | Ret. Time | Name | Flight class | Price | Connection |
| val. | mor/aft/eve/all | mor/aft/eve/all | cate. | eco/bus/all | 500-5000 | 0/1/all |
| feature | Airline | | | | | |
| val | norm/all | | | | | |

needs to learn to generalize between those concepts in order to succeed. To limit the amount of database data to be generated for each sample, we only generate 30 candidate flights for each dialogue. Both user restrictions and database are generated according to a preset prior. For example, we limit the percentage of people who requires business class to be 1%. Those priors ensure that we can have relatively high booking rate even when the number of flights in the database is small. We have observed that 55.34% of the user restrictions resulted in a book status while 24.68% of of them results in a flight not found state. In addition to the flight database, we also generate a single entry indicating whether a user has reservation in the system.

**Dialogue and Action Generator.** The dialogue data is generated according to Figure 2 based on a generated pair of user restrictions and database. To initialize the conversation, we randomly pick a speaker to start. We divided the conversation into stages and at each stage we have prepared several different language templates. They are randomly chosen when the dialogue is being generated. Actions are generated diagnostically based on the dialogue setups. They contain three fields: the first field contains the final state of the dialogue (i.e., book,no flight, no reservation, cancel, changed). The second field contains the name of the person. The third field contains the flight number to be recommended. In the case of canceling a flight, the we output a special token "<flight_empty>". Both the action and the dialogue generated by the simulator are guaranteed follow the optimal policy.

**Reward Generator.** Rewards are generated diagnostically based on predicted action and the true action as well as the predicted dialogue. To generate rewards, we first check several language rules

Table 3: Features of Flight Databases

| no. | 1 | 2 | 3 | 4 | 5 | 6 | 7 |
|---|---|---|---|---|---|---|---|
| feat. | Dep. City | Ret. City | Dep. Month | Ret. Month | Dep. Day | Ret. Day | Dep. Time |
| val. | cate. | cate. | 1-12 | 1-12 | 1-31 | 1-31 | 00-24 |
| feature | Ret. Time | Flight class | Price | Connection | Airline | flight no. | |
| val. | 00-24 | eco/bus | 0-5000 | 0/1/2 | cate. | cate. | |

Table 4: Experimental Results - Evaluations on Supervised Training

| Experiments | Eval ppl | Eval BLEU | Eval Reward | Eval name | Eval Flight | Eval Action |
|---|---|---|---|---|---|---|
| Supervised | 1.214 | 33.6 | 1 | 0.948 | 1.00 | 1.00 |

Table 5: Experimental Results - Evaluations on Self-play

| Experiments | self-play eval reward |
|---|---|
| Supervised | 0.321 |
| Self-play | 0.441 |

on the predicted dialogue. Those rules include things such as dialogue must contain end of dialogue token or dialogue must not have consecutive repeating tokens. If those language rule check fails, reward will be zero. These discourages the model from generating very back utterances that might benefit the performance in short term but will hurt in the long term. We then check if the predicted action state is exactly the same as the ground truth state. If not the reward will be zero. In the final case, we check whether the name matches the correct action, which will counts for 50% of the final reward. The other 50% goes to database recommendation checks, where we calculates a squared error on each dimension of the flight and normalized the distance against the optimal one. This setup of generating rewards discourages wrong action state and invalid dialogues strongly, which helped to generate dialogues that are of high quality.

## 4 EXPERIMENTS

**Implementation Details.** We experimented our supervised model using 8 Nvidia Tesla P100 GPUs on a single machine. In the self-play we utilized 30 machines each with a Tesla P100 using distributed training. During self-play we conduct 1 supervised training using the training data every time we make a reinforcement update to avoid the our model from deviating human language. We use 4 layers of GRU in utterance decoding and encoding with 512 units. The value network is chosen to be 2 layer GRU with 256 units.

**Numerical Results.** To evaluate our models, we first trained our supervised model based on the training dataset and evaluated it on the held-out evaluation set. The results are illustrated in Table 4. As we can see from the table, the model trained using supervised learning has low perplexity and satisfactory BLEU score on the evaluation dataset. Note that although this is a simulated dataset, the fact that each sentence has more than three random possibilities prohibited it to achieve perfect BLEU scores. From the table we can also observe that the model achieved a perfect score on supervised setting, topping the accuracy of three actions as well as the reward. Since this is the supervised setting, the action predictions are based on perfect dialogues, its success won't transfer to the self-play evaluation case. As we have seen from Table 4, the performance of the supervised model drop significantly in the self-play evaluation. Using self-play, however, we have improved the reward for over 37%. Although this number is far below the perfect score, the improvement of self-play is obvious.

**Qualitative Results.** To see how our model performs qualitative, here are two examples generated using the self-play version of the model. In the first example, the customer asked to book a flight from MCO to DTW. The service agent gradually asked for information and they were able to locate the flight that satisfies all customer's requests. In the second example, the customer asks to change a reservation. However, the service agent did not find the reservation in the system and the request can not be fulfilled.

Sample Dialogue 1
<t1> Hello .
<t2> Hi .
<t1> I want to book a flight .
<t2> Not a problem . What 's your departure and arrival city ?
<t1> I will be flying from MCO to DTW .
<t2> What 's the departure and return date ?

<t1> I will be departing on Mar 20 and returning on Mar 22 .
<t2> We have flight 1011 departs from PHX on Mar 20 at 12:00 in the afternoon and returns from DTW on Jan 22 at 14:00 in the afternoon . This is a economy class flight with 1 connection . The total price is $100 .
<t1> Looking good .
<t2> What 's the name for this reservation ?
<t1> My name is Matthew_Hill .
<t2> I just want to confirm with your on flight 1011 . Is this the ticket we are going to book ?
<t1> confirmed . Please go ahead .
<t2> We have successfully booked your trip . Thanks for using our flight booking service . <eod><t1>

Sample Dialogue 1
<t1> Hello .
<t2> Hi .
<t1> I need helps to change my reservation .
<t2> I am sorry , but I can not locate your reservation . <eod> <t1>

## 5 RELATED WORK

The dominant paradigm in dialogue modeling is using hand crafted rules and statistical models. Examples include Markov Decision Process (MDP) (Levin et al., 1997; 2000), POMDP (Gasic et al., 2013) and statistically tuned rule based models (Oh & Rudnicky, 2000). Another approach of studying dialogue system is to divide the problems into several smaller problems, such as the dialogue state tracking challenge (Williams et al., 2013) and the bAbI tasks (Weston et al., 2015; Bordes & Weston, 2016). The introduction of the seq2seq modeling (Sutskever et al., 2014) and the idea of transforming the dialogue problem into a translation problem (Ritter et al., 2011) rapidly transformed the field to utilize neural based learning techniques (Vinyals & Le, 2015; Wen et al., 2015; Shang et al., 2015; Li et al., 2016a; Luan et al., 2016). Many recent works leverage deep reinforcement learning to either promote the diversity of the conversation (Li et al., 2016b) or to apply it on goal oriented conversation models (Li et al., 2016c). Lewis et al. (2017) managed to build a conversation model based on self-play. However, their models are shared across two agents and their tasks to generate negation items can easily degenerate into classification problems without the need to interact through dialogue. Another line of work combines dialogue with problems in other domains such as vision problems (Morguet & Lang, 1999).

## 6 CONCLUSION

In this paper, we proposed a new approach to model goal-oriented conversations based on information isolation and action states. By leveraging supervised learning with self-plays on actions states, we expanded the coverage of the training significantly and exposed the model with unseen data. By enforcing information isolation, we tightly coupled dialogue data with action states. Results indicated self-play under those settings significantly improved the reward function compares to the supervised learning baseline. Since dialogue data is usually hard to get while action states can be acquired easily, our approach can be easily applied in those scenario where the amount of the data is a bottleneck to the performance of the system.

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
