# OpenReview forum: "A Goal-oriented Neural Conversation Model by Self-Play"
_ICLR.cc/2018/Conference — Reject_

### Official Review · AnonReviewer2 · 2017-11-22
**An interesting way to force the dialogue model to be closer to the conversational one, via sharing hidden information. The paper is not clear is more part and human evaluation is missing.**

**Rating:** 6
**Confidence:** 3

**Review:**

I like the idea of coupling the language and the conversation model. This is in line with the latest trends of constructing end-to-end NN models that deal with the conversation in a holistic manner. The idea of enforcing information isolation is brilliant. Creating hidden information and allowing the two-party model to learn through self-play is a very interesting approach and the results seem promising.

Having said that, I feel important references are missing and specific statements of the paper, like that "Their success is however limited to conversations with very few turns and without goals" can be argued. There are papers that are goal oriented and have many turns. I will just provide one example, to avoid being overwhelming, although more can be found in the literature. That would be the paper of T.-H. Wen, D. Vandyke, N. Mrksic, M. Gasic, L. Rojas-Barahona, P.-H. Su, S. Ultes and S. Young (2017). "A Network-based End-to-End Trainable Task-oriented Dialogue System." EACL 2017, Valencia, Spain. In fact in this paper even more dialogue modules are coupled. So, the "fresh challenge" of the paper can be argued.

It is not clear to me how you did the supervised part of the training. To my experience, although supervised learning can be used, reinforcement learning seems to be the most popular choice. Also, I had to read most of the paper to understand that the system is based on a simulator. Additionally, it is not clear how you got the ground-truth for the training. How are the action and the dialogue generated by the simulator guaranteed to follow the optimal policy?

I also disagree with the statement that "based on those... to estimate rewards". If ruled-based systems were sufficient, there would not be a need for statistical dialogue managers. However, the latter is a very active research area.

Figure 1 is missing information (for my likings), like not defined symbols. In addition, it's not self-contained. Also, I would prefer a longer, descriptive and informative label to make the figure as self-explained as possible. I believe it would add to the clarity of the paper.

Also, fundamental information, according to my opinion is missing. For example, what are the restrictions R and how is the database K formed? What is the size of the database? How many actions do you define? Some of them are defined in the action state decoder, but it is not clear if it is all of them.


GRU -> abbreviation not defined

I would really appreciate a figure to better explain the subsection "Encoding External Knowledge". In the current form I am struggling to understand what the authors mean.

How is the embedding matrix E created?

Have you tried different unit sizes d? Have you tried different unit sizes for the customer and the service?

"we use 2 transformation matrixes" -> if you could please provide more details

How is equation 2 related to figure 1?

Typo: "name of he person"

"During the supervised learning... and the action states". I am not sure I get what you mean. May you make this statement more clearly by adding an equation for example?

What happens if you use random rather than supervised learning weight initialisation?

Equation 7: What does T stand for?

I cannot find Table 1, 2 and 5 referred to in-text. Moreover I am not sure about quite some items. For example, what is number db? What is the inference set?

500k of data is quite some. A figure on convergence would be nice.

Setting generator: You mention the percentage of book and flight not found. What about the rest of the cases?


Typo: “table 3 and table 3”

The set of the final states of the dialogue is not the same as the ones presented at Fig. 2.

Sub section reward generation is poorly described. After all, reward seems to play a very important role for the proposed system. Statements like  “things such as” (instead the exhaustive list of rules for example) or “the error against the optimal distance” with no note what should be considered the optimal distance make the paper clarity decreased and the results not possible to be reproduced. Personally I would prefer to see some equations or a flow chart. By the way, have you tried and alternative reward function?


Table 4 is not easy for me to understand. For example, what do you mean when you say eval reward?

Implementation details. I fail to understand how the supervised learning is used (as said already). Also you make a note for the value network, but not for the policy network.

There are some minor issues with the references such as pomdp or lstm not being capitalised


In general, I believe that the paper has a great potential and is a noticeable work. However,
the paper could be better organised. Personally, I struggled with the clarity of some text portions.
For me, the main drawback of the paper is that it was't tested with human users. The actual success of the system when evaluated by humans can be surprisingly different from the one that comes from simulation.

---

### Official Review · AnonReviewer1 · 2017-11-27
**Contribution is not clear w.r.t. previous work**

**Rating:** 4
**Confidence:** 3

**Review:**

Summary: The paper proposes a self-play model for goal oriented dialog generation, aiming to enforce a stronger coupling between the task reward and the language model.

Contributions:

While there are architectural changes (e.g. the customer agent and client agent have different roles and parameters; the parameters of both agents are updated via self-play training), the information isolation claim is not clear. Both the previous work (Lewis et al., 2017) and the proposed approach pitch two agents against each other and the agents communicate via language utterances alone (e.g. rather than exchanging hidden states). In the previous work, the two agents share a set of initial conditions (the set of objects to be divided; this is required by the nature of the task: negotiation), but the goals of each agent are hidden and the negotiation process and outcome are only revealed through natural language. Could you expand on your claim regarding information isolation? Could you design an experiment which highlights the contribution and provide a comparison with the previous approach?

Furthermore, divergence from natural language when optimizing the task reward remains an issue. As a result, both methods require alternate training between the supervised loss and the reinforcement loss.

Experiments:

1. Minor question: During self-play "we conduct 1 supervised training using the training data every time we make a reinforcement update". One iteration or one epoch of supervised training?

2. The method is only evaluated on a toy dataset where both the structure of the dialog is limited (see figure 2) and the sentences themselves (the number of language templates is not provided). The referenced negotiation paper uses data collected from mechanical turk ensuring more diversity and the dataset is publicly available. Couldn't your method be applied to that setting for comparison?

3. The qualitative evaluation shows compelling examples from the model. Are the results hand-picked to highlight the two outcomes? I wish more examples and some statistics regarding the diversity of produced dialogs were provided (e.g. how many times to they result in a booked flight vs. unfulfilled request and compare that with the training data).

4. What is the difference between evaluation reward reported in Table 4 and self-play evaluation reward reported in Table 5? (Is the former obtained by conditioning on target utterances?). Is there a reason to not report the itemized rewards in Table 5 as well (Eval flight, Eval action) etc?

5. The use of the value network vs. the policy network is not clarified in the model description nor in the experiments. Is the value network used to reduce the variance in the reward?

Finally, there are several typos or grammatical errors, including:
- Page 4, t and i should be the same.
- Page 4. Use p(u_t |  t_{<t-1}; \theta) instead of p(u_t |  t_{<t-1} | \theta).
- Page 2, second paragraph: "which correctness" -> "whose correctness".
- Page 2, second-to-last paragraph: "access to a pieces" -> "access to pieces", "to the best of it can" -> "as good as it can".
- Page 4. "feeded" -> fed
- Page 5, second-to-last paragraph: "dataset is consists of" -> "dataset consists of".
- Page 7/8: Both examples are labeled "Sample dialog 1"
- Dataset & experiments: Table 3 and Table 3
- Experiments: "to see how our model performs qualitative" -> "to see how our model performs qualitatively"
- Related work: "... of studying dialog system is to ..." -> "dialog systems"
- Conclusion: "In those scenario" -> "In those scenarios"

---

### Official Review · AnonReviewer3 · 2017-11-27
**Unsure what the contribution is**

**Rating:** 3
**Confidence:** 4

**Review:**

This paper describes a method for improving a goal oriented dialogue system using selfplay. Using similar techniques to previous work, they pretrain the model using supervised learning, and then update it with selfplay reinforcement learning. The model is evaluated on a synthetic flight booking task, and selfplay training improves the results.

I found it very difficult to work out what the contribution of this paper is over previous work, particularly the Lewis et al. (2017) paper that they cite. The approach to using selfplay RL seems almost identical in each case, so there doesn’t appear to be a technical contribution. The authors say that in contrast to Lewis et al. their “setting enforces the isolation of information between the models of the two agents” - I’m not sure what this means, but the agents in Lewis et al. each have information that the other does not. They also claim that Lewis et al.’s task “can easily degenerate into classification problems without the need to interact through dialogue”, but offer no justification for this claim.

The use of a synthetic dataset also weakens the paper, and means it is unclear if the results will generalize to real language. For example, Lewis et al. note challengings in avoiding divergence from human language during selfplay, which are likely to be less pronounced on a synthetic dataset.

Overall, the paper needs to be much clearer about what its contributions are over previous work before it can be accepted.

---

### Decision · Program_Chairs · 2018-01-29
**ICLR 2018 Conference Acceptance Decision**

**Decision:**

Reject

**Comment:**

While using self-play for training a goal-oriented dialogue system makes sense, the contribution of this paper compared to previous work (that the paper itself cites) seems too minor, and the limitations of using toy synthetic data further weaken the work.